# Interplay of the forces governing steroid hormone micropollutant adsorption in vertically-aligned carbon nanotube membrane nanopores

Minh N. Nguyen [1], Melinda L. Jue [2], Steven F. Buchsbaum[2], Sei Jin Park [2], Florian Vollnhals [3], Silke Christiansen[3,4], Francesco Fornasiero [2] & Andrea I. Schäfer [1] ✉

Vertically-aligned carbon nanotube (VaCNT) membranes allow water to conduct rapidly at low pressures and open up the possibility for water purification and desalination, although the ultralow viscous stress in hydrophobic and low-tortuosity nanopores prevents surface interactions with contaminants. In this experimental investigation, steroid hormone micropollutant adsorption by VaCNT membranes is quantified and explained via the interplay of the hydrodynamic drag and friction forces acting on the hormone, and the adhesive and repulsive forces between the hormone and the inner carbon nanotube wall. It is concluded that a drag force above $2.2 \times 10^{-3}$ pN overcomes the friction force resulting in insignificant adsorption, whereas lowering the drag force from $2.2 \times 10^{-3}$ to $4.3 \times 10^{-4}$ pN increases the adsorbed mass of hormones from zero to 0.4 ng cm$^{-2}$. At a low drag force of $1.6 \times 10^{-3}$ pN, the adsorbed mass of four hormones is correlated with the hormone−wall adhesive (van der Waals) force. These findings explain micropollutant adsorption in nanopores via the forces acting on the micropollutant along and perpendicular to the flow, which can be exploited for selectivity.

Water scarcity on a global scale constitutes a monumental challenge[1,2] to accomplishing the Sustainable Development Goals set by the United Nations[3]. Four billion people do not have access to freshwater (surface water and groundwater) for at least one month in a year, and half a billion people suffer from water scarcity all year round[4]. Severe water scarcity occurs in the Middle East, Southeast Asia, and especially Africa, where up to 40% of the population has limited or no access to safe water[5]. To alleviate this water stress, it is a priority to seek and use alternative water sources, such as reclaimed wastewater and seawater[1,6]. A core technology of both water reuse and desalination is reverse osmosis (RO) that is capable of removing non-selectively various organic pollutants, and multi- and monovalent ions[7]. RO requires pressures of 5–80 bar[7] (depending on the osmotic pressure) and specific energy consumption of 2.3–5.2 kW h m$^{-3}$ for seawater desalination[8] and 0.4–1.7 kW h m$^{-3}$ for brackish water desalination and water reuse[9]. Exclusively for water reuse, nanofiltration (NF) combines operation at 3−20 bar pressures[7] (corresponding to a specific energy consumption range[10,11] of 0.2−0.5 kW h m$^{-3}$) and good separation of organic pollutants[12,13], including micropollutants occurring at sub-nanogram- or several-microgram-per-litre concentrations[14].

The separation capability of the membrane is indicated by the molecular-weight cut-off (MWCO)[15], which is the minimum molecular

[1]Institute for Advanced Membrane Technology (IAMT), Karlsruhe Institute of Technology (KIT), Eggenstein-Leopoldshafen, Germany. [2]Physical and Life Sciences Directorate, Lawrence Livermore National Laboratory (LLNL), Livermore, CA, US. [3]Institute for Nanotechnology and Correlative Microscopy (INAM), Forchheim, Germany. [4]Fraunhofer Institute for Ceramic Technologies and Systems (IKTS), Forchheim, Germany. ✉e-mail: andrea.iris.schaefer@kit.edu

weight of the solute that results in 90% retention. However, NF/RO membranes cannot remove micropollutants completely even when these compounds have higher molecular weights than the membrane MWCOs[16,17] due to the presence of larger pores and defects[18]. For instance, steroid hormone micropollutants (around 300 Da) are important removal targets because these compounds may disrupt the functions of the endocrine system in the body[19,20]. They occur at up to a few hundred nanogram-per-litre concentrations in wastewater effluents[21,22]. Typical NF membranes such as the loose NF270 and dense NF90 (DuPont, USA) only remove 70–90% of uncharged steroid hormones from a feed concentration of 100 ng L$^{-1}$ at neutral pH[23–25]. A 99% removal has not been attained although this level is required to achieve the very low guideline concentration in drinking water (1 ng L$^{-1}$) proposed by the European Union for 17β-estradiol (E2)[26]. The low removal results from the adsorption of steroid hormones to the membrane, and subsequent diffusion through the membrane materials[17,27]. A steroid hormone *breakthrough* is detected as a consequence, where the permeate concentration increases over time until when the membrane polymer is saturated with the adsorbed molecules[23,25]. Because adsorption to the membrane leads to subsequent transport through the pores and hence lower removal, a desirable highly-selective membrane needs to display not only good retention of pollutants but also low adsorption capability toward the pollutants.

The structural non-uniformity of state-of-the-art NF/RO membranes presents challenges in investigating the adsorption behaviour in membrane materials and nanopores. These membranes consist of a thin-film polyamide layer (50–200 nm in thickness) formed by interfacial polymerisation, a microporous support layer (~50 μm), and a nonwoven substrate (~200 μm)[28]. The thin polyamide layer that governs the transport of water and solutes is characterised by the varying morphology and chemical composition along the layer thickness[29–31]. Molecular dynamics simulations suggest that the interactions between water and the functional groups in the polyamide layer pose significant resistance to water transport[32,33]. Using scanning transmission electron microscopy with high-angle annular dark-field imaging (HAADF-STEM), Culp et al. mapped the spatial density of the polyamide layer in three dimensions and revealed that water and solute molecules might selectively permeate through the paths of lowest resistance (*i.e.* lowest layer thickness and density or defects)[34]. This phenomenon results in a reduced number of actual transport channels, and makes it difficult to quantify the extent of membrane surface interactions with the solutes.

Unlike the conventional NF/RO membranes, advanced membranes designed from the molecular level, including the vertically-aligned carbon nanotube (VaCNT) membranes[35], are ideal model systems for investigating adsorption and breakthrough in nanopores. In VaCNT membranes, the carbon nanotubes orientate vertically forming the membrane pores with well-defined geometrical structure (cylinder), low tortuosity, and good chemical and structural uniformity along their entire length. With a pore diameter of ~2 nm and porosities of 0.6–4.3%, VaCNT membranes have water permeabilities of 70–300 L m$^{-2}$ h$^{-1}$ bar$^{-1}$ [36,37], which are higher than those of ultrafiltration membranes (UF) with similar pore diameters and higher porosities of 5–11% (5–9 L m$^{-2}$ h$^{-1}$ bar$^{-1}$)[38], and the entire permeability range for NF membranes with similar or higher porosities of 2–32% (4–17 L m$^{-2}$ h$^{-1}$ bar$^{-1}$)[24]. VaCNT membranes also enable good separation and selective transport of solute molecules and ions[37,39]. The selectivity can be modulated by tuning the pore diameter[40,41] and via tip functionalisation[42,43].

In nanopores where the sizes of solutes or water are only several times lower or even comparable to the pore size, transport phenomena are primary governed by the interactions between these solutes or water and the pore surface of the membrane[44]. Water permeation in NF/UF membranes is suppressed by a high viscous stress[45] on the hydrophilic pore 'wall'[7]. The Poiseuille flow with a no-slip boundary

condition (where the fluid at the boundary have zero velocity) applies loosely to these membranes, and leads to a parabolic flow velocity profile in cylindrical pores and zero velocity at the wall[7,46] (Fig. 1A). The active layer pores of asymmetric NF/UF membranes are tortuous and not cylindrical[47], and Poiseuille flow may be invalid for <1 nm NF membrane pores as the continuum hydrodynamics breaks down[48]. The Hagen−Poiseuille equation is used to determine the flow rate $Q_{HP}$ (m$^3$ s$^{-1}$) in a UF/NF membrane pore (Eq. (1)).

$$Q_{HP} = \frac{\pi \Delta P}{8\mu\alpha L}\left(\frac{d_p}{2}\right)^4 \qquad (1)$$

where $\Delta P$ (Pa) is the pressure drop across the length of the pore, $\mu$ (Pa.s) is the water viscosity, $d_p$ and $L$ (m) are the pore diameter and length, and $\alpha$ is the tortuosity factor ($\alpha = 1$ for non-tortuous and >1 for tortuous pores[15]; foam-like pores deviate strongly from this relationship). High $\alpha$ values of UF/NF membranes (1.5–2.5[47]) cause more water to collide with the pore surface and adds resistance to the flow[49,50].

The no-slip boundary condition in the Poiseuille flow cannot explain the anomalously high water conduction in VaCNT membranes compared with UF/NF membranes[51]. This flow enhancement is attributed to pore surface hydrophobicity, smoothness, and low pore tortuosity. Hydrophobic graphene-like surfaces permit water slip where the water moves at a non-zero velocity at the wall (which is called the slip velocity) because of the low fluid−wall viscous stress[52] (Fig. 1B, C). Some hydrophilic surfaces may cause a small slip in theory, although this has not been confirmed experimentally[53]. In carbon nanotubes with >1 nm diameter pores, slip is partly explained via the shallow potential energy landscape[54], low quantity of electrons at the surface[55], electronic oscillation of the nanotube[56], and quantum-level coupling between oscillating nanotubes and fluctuating water charges[57]. Unlike the rough UF/NF membrane pore surfaces resulting from an interconnected pore network[58,59], those of VaCNT membranes are atomically smooth[54]. Any increase in pore surface roughness by 0.3 nm (*i.e.* the size of a single water molecule) leads to diffusive scattering, promotes water−wall collisions, and diminishes the fast flow[60,61]. On an atomically smooth surface, the 0.3 nm gap, which is the electron cloud thickness between the nuclei of the confined water molecules and those of the carbon nanotube wall[62], is an important separation distance for maintaining low viscous stress and high slip.

With the flow enhancement $Q_{en}$ defined as $\frac{\pi\Delta P}{8\mu L}\left[4b\left(\frac{d_p}{2}\right)^3\right]$, in which the slip is quantified by the slip length $b$ (m)[45,63], the slip-corrected flow rate $Q_{slip}$ in a nanopore is given by Eq. (2)[36]; $\alpha$ is low (1.1–1.25[64,65]) and can be set to 1.

$$Q_{slip} = Q_{en} + Q_{HP} = \frac{\pi\Delta P}{8\mu\alpha L}\left[\left(\frac{d_p}{2}\right)^4 + 4b\left(\frac{d_p}{2}\right)^3\right] \qquad (2)$$

In VaCNT membranes, the slip length $b$ is very high (*i.e.* $b\gg\frac{d_p}{2}$) resulting in 3−4 orders of magnitude of enhancement factor (determined as the ratio between $Q_{slip}$ and hypothetical $Q_{HP}$)[36,51,64,66]. With very high slip lengths and enhancement factors, the flow velocity at the wall is close to that in the pore centre, and the plug flow regime (Fig. 1D) can be assumed[67,68].

In a pressure-driven process with the VaCNT membrane or UF/NF, an uncharged solute, such as a steroid hormone molecule, can interact more strongly with the pore wall and hence move more slowly than water. If the movement of the hormone is too slow, the hormone appears adsorbed by the pore wall. The significance of this 'adsorption' is the result of the forces acting on the steroid hormone, which are the 1) hydrodynamic drag force $F_H$ that drives the hormone movement in the flow direction, 2) adhesive force $F_A$ directed at the pore wall that results in adsorption, 3) repulsive force $F_R$ that balances $F_A$ and keeps the hormone molecule at the wall, and 4) hormone−wall friction force $F_F$ that resists the hormone movement with the flow and opposes $F_H$

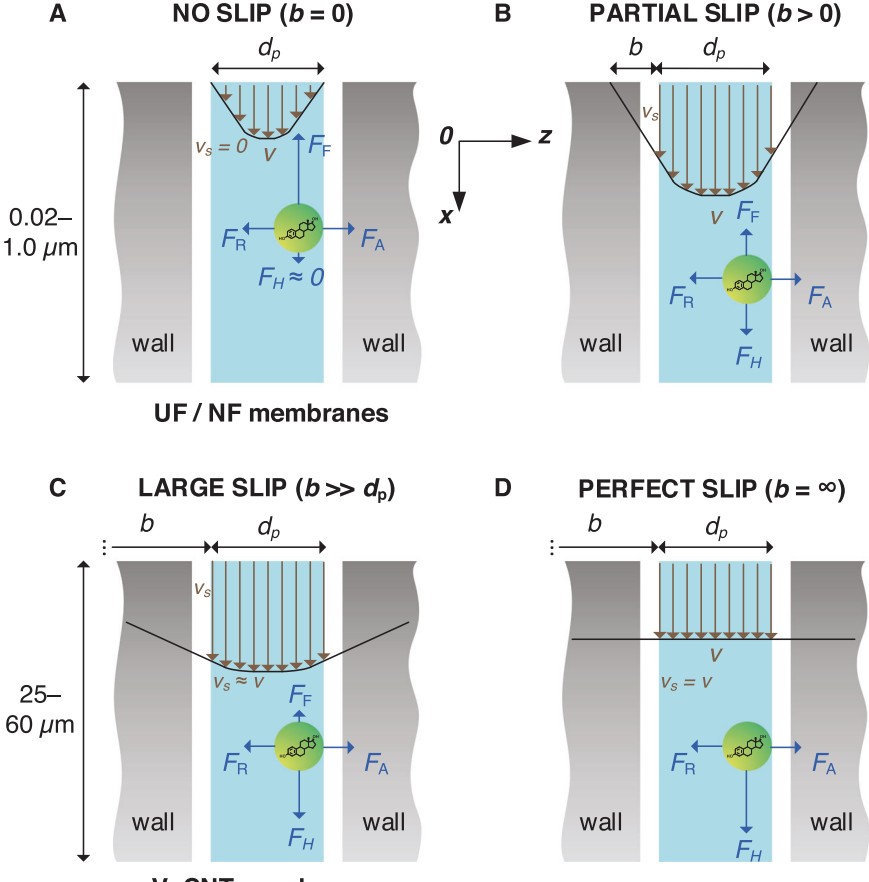

**Fig. 1 | Flow in nanopores with diameter $d_p = 1.7$ nm. A** no slip (slip length $b = 0$) in UF/NF membranes. **B** partial slip ($b > 0$). **C** large slip ($b \gg d_p$) in VaCNT membranes. **D** (hypothetical) perfect slip ($b = \infty$). A 17β-estradiol (E2) molecule (green sphere, with a hydrodynamic diameter of 0.8 nm) at the liquid−wall interface under the influence of adhesive force $F_A$, its countering repulsive force $F_R$, hydrodynamic drag force $F_H$, and hormone−wall friction force $F_F$. The pore and E2 diameters are approximately to scale.

(Fig. 1). The significance of these forces depend on the material (molecule and surface) properties (Supplementary Table 1).

In VaCNT membranes, the viscous stress and shear forces applied by the water flow on the steroid hormone molecule (assumed to be spherical with hydrodynamic diameter $d_{SH}$) result in an $F_H$ that can scale proportionally with the water flow velocity $v_{water}$ (m s$^{-1}$) as shown in Eq. (3)[69]. A plug-like unbound flow condition[70] is assumed as an approximation of the real profile, so $v_{water}$ at the wall is equal to that in the pore centre.

$$F_H = 3\pi \mu d_{SH} v_{water} \qquad (3)$$

The van der Waals interaction[71] is assumed to be responsible for $F_A$ is related to the hormone diameter $d_{SH}$, hormone−wall distance and intrinsic affinity via the Hamaker constant $H$ (J) specific for the pair of steroid hormone and pore wall[72,73] (see Eq. (4)).

$$F_A = \frac{H d_{SH}^3}{16\left(z - \frac{1}{2}d_{SH}\right)^4} \qquad (4)$$

where $z$ (m) is the distance between the steroid hormone molecule centre and the pore wall. $H$ is assumed to be independent of the geometries of the interaction species; in reality, specific groups (namely the π-rings) of the hormone and pore wall may interact more strongly[74]. When $z - \frac{1}{2}d_{SH} \approx 0.3$ nm (which is the thickness of the electron cloud between the nanotube wall and molecules at the fluid

−wall interface), $F_A$ balances the repulsive force $F_R$ between the electrons of the pore wall and those of the steroid hormone molecule (an illustration of this force balance through Lennard−Jones-type potential is given in Supplementary Fig. 1), keeping the hormone molecule static in the direction perpendicular to the flow[73].

In dynamic adsorption studies, adsorption surface and mass transfer are deemed the key limiting factors to adsorption[75,76]. However, the interplay of the forces ($F_H$, $F_F$, $F_A$, and $F_R$) may play an important role inside the membrane nanopores where both surface and mass transfer are no longer the limiting factors, as steroid hormone molecules readily access the nanopore surface. In thin-film composite NF membranes, the adsorbed mass of steroid hormones is 0.2−1.5 ng cm$^{-2}$ with adsorption being dominant in the polyamide active layer[24]. The hormone−wall friction $F_F$ is high, depending not only on the intrinsic hormone−wall interaction[77], but also on the pore surface roughness and pore tortuosity[47] that are large in NF membrane pores. With the (inaccurate) assumptions of the Poiseuille flow and no-slip condition[7], the drag force $F_H$ is weak because of the low water velocity close to the wall. Strong $F_F$ and weak $F_H$ in NF membranes would result in significant adsorption. In VaCNT membranes, strong $F_H$ due to the fast water flow velocity at the wall and weak $F_F$ resulting from slippage may drive the steroid hormone molecules to exit the pores. A threshold of $F_H$ can be quantified, below which adsorption becomes significant.

The main focus of this study is to investigate how the interplay of forces affects adsorption in the VaCNT membrane and to extend these findings to other nanopores. Specifically, water filtration and steroid

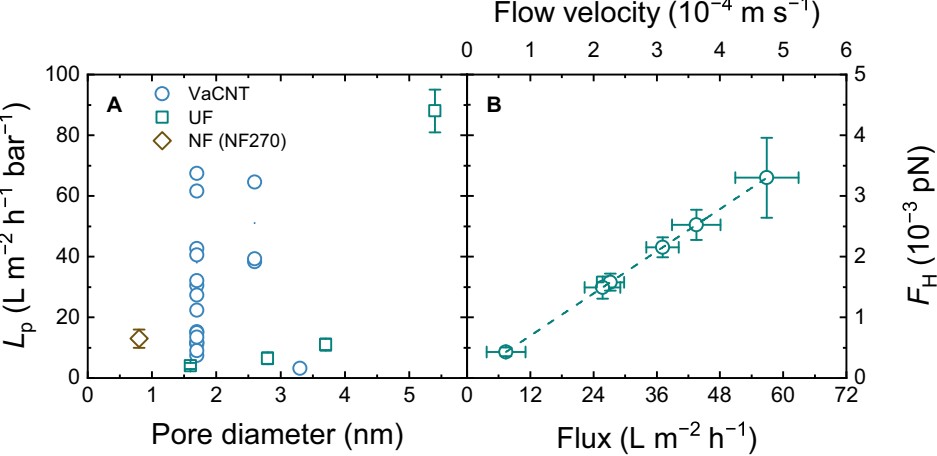

**Fig. 2 | Filtration property of VaCNT membranes. A** Pure water permeability $L_p$ of VaCNT benchmarked against UF and NF270 membranes. **B** Linear relationship between flux, flow velocity in the pores, and hydrodynamic drag force $F_H$ acting on an E2 molecule specific for the 1.7 nm pore diameter VaCNT membranes. Error bars represent propagated error from operational parameter variations.

hormone adsorption properties were examined in the slippery and structurally uniform VaCNT pores and benchmarked against the much more heterogeneous and tortuous pores of commercial UF/NF membranes. Then, the influence of flow hydrodynamics and hormone−wall affinity on the adsorption by VaCNT membranes was assessed by quantifying the respective hydrodynamic drag force $F_H$ and adhesive force $F_A$. Finally, the proposed force interplay framework is applied to explain steroid hormone adsorption in various types of nanopores.

## Results and Discussion

### Water permeability in the nanopores

Good water conduction resulting from large slippage in VaCNT membranes is a prerequisite for achieving a strong drag force at the fluid−wall interface. This drag force is the main driving force for adsorbed steroid hormone molecules to move along and exit the pores. The pure water permeability of VaCNT membranes (pore diameters of 1.7−3.3 nm) is reported and compared with commercial UF (pore diameters 1.6−5.4 nm, PL series, Millipore)[38,76,78] and NF membranes (NF270, Dupont, pore diameters 0.8 nm)[79] in Fig. 2A. The same micro-crossflow filtration system was used for this comparison[79]. The properties of the UF/NF membranes are given in Supplementary Table 2. The VaCNT membrane was placed on a microfiltration (MF) support to reduce the stress on the VaCNT layer during filtration. The MF support has a permeability >500 L m$^{-2}$ h$^{-1}$ bar$^{-1}$, much larger than that of the VACNT membrane), so that the measured permeability of the VACNT−MF pair is effectively equal to the permeability of VaCNT.

From Fig. 2A, at first glance, the permeabilities of VaCNT membranes were not significantly higher than those of UF/NF membranes. With pore diameters of 1.7 and 2.6 nm, the permeabilities of VaCNT membranes varied between 8 and 70 L m$^{-2}$ h$^{-1}$ bar$^{-1}$, which are in the same range as or an order of magnitude higher than those of the NF270, and in the same range as UF with pore diameters of 1.6−3.7 nm. The 3.3 nm diameter VaCNT membrane had a slightly lower permeability than the UF with pore diameters of 2.8−3.7 nm. From a rich set of literature data, Mattia et al. reported that many VaCNT membranes had water permeabilities that are only zero or one order of magnitude higher than the corresponding UF/NF membranes[51], which agrees with the findings of this work. The VaCNT membrane can hence be operated at typical fluxes of 15−100 L m$^{-2}$ h$^{-1}$ in UF with pore diameters below 6 nm[38,78] and 40−200 L m$^{-2}$ h$^{-1}$ in NF membranes[25]. With a nominal porosity (3.4%) in the low end of the porosity range of UF (1−15%[80]) and NF membranes (2−32%[24]) the VaCNT membrane enables higher flow velocity than UF/NF membranes at the same flux.

From Supplementary Fig. 2 and Supplementary Table 3, the enhancement factors (720−6200) and slip lengths (75−650 nm) for the 1.7 nm diameter VaCNT membranes were in a similar range as reported by Holt et al.[36]. The VaCNT membranes with pore diameters of 2.6 and 3.3 nm also gave high enhancement factors of 690−1200 and 174, respectively. The several-orders-of-magnitude flow enhancement in VaCNT compared with the no-slip Hagen−Poiseuille does not translate to a permeability gain of similar orders of magnitude with respect to commercial UF/NF membranes. This is due to the comparably low porosity and much larger thicknesses of VaCNT membranes. The large variation in enhancement factors and slip lengths is caused by the rough (over)estimation of the number of conducting carbon nanotubes (all nanotubes in the membranes are assumed to conduct fluids), and contribution of neglected entrance resistances. According to molecular dynamics simulations, the significance of entrance (and exit) resistances are limited to a certain carbon nanotube length (below 20 μm), as water flowing through longer nanotubes experiences considerable resistance due to viscous friction along the walls[81,82]. Secchi et al. quantified the slip lengths of short (0.45−1 μm) strands of individual carbon nanotubes with large internal diameters (30−100 nm) after subtracting the entrance resistances[83] and reported values similar to those of the VaCNT membranes with 10 times smaller pore diameters[37,84].

In all water permeability experiments, the slip length was very high (i.e. $b \gg \frac{d_p}{2}$ where $d_p$ is the pore diameter), and the enhancement factors are 2−4 orders of magnitude (see Supplementary Fig. 2 and Supplementary Table 3). As such, a plug-like flow condition can be assumed for water[62,63]. Under this assumption, if a steroid hormone molecule such as 17β-estradiol (E2) is introduced inside the VaCNT membrane pore, the hydrodynamic drag force $F_H$ acting on the hormone molecule will increase proportionally with the flow velocity via Eq. (3), even when the hormone is at the fluid−wall interface. Figure 2B presents the dependence of $F_H$ on the water flow velocity and flux. With an increase in flux from 6 to 60 L m$^{-2}$ h$^{-1}$, the flow velocity increases from 0.06 to 0.47 mm s$^{-1}$, and $F_H$ increases from $4.2 \times 10^{-4}$ to $3.5 \times 10^{-3}$ pN. The Péclet number increases from 1.0 to 11.3 with the increasing flux from 6 to 60 L m$^{-2}$ h$^{-1}$ assuming a diffusivity of water in the pores similar to that of the bulk ($2.3 \times 10^{-9}$ m$^2$ s$^{-1}$)[85]. The Péclet number is greater than 1 except for the case of the lowest flux of 6 L m$^{-2}$ h$^{-1}$ (when the Péclet number is equal to 1), which indicates that the transport is dominated by advection and justifies the analysis of the hydrodynamic drag force $F_H$. With the lowest flux of 6 L m$^{-2}$ h$^{-1}$, both advection and diffusion influenced the water transport, and the

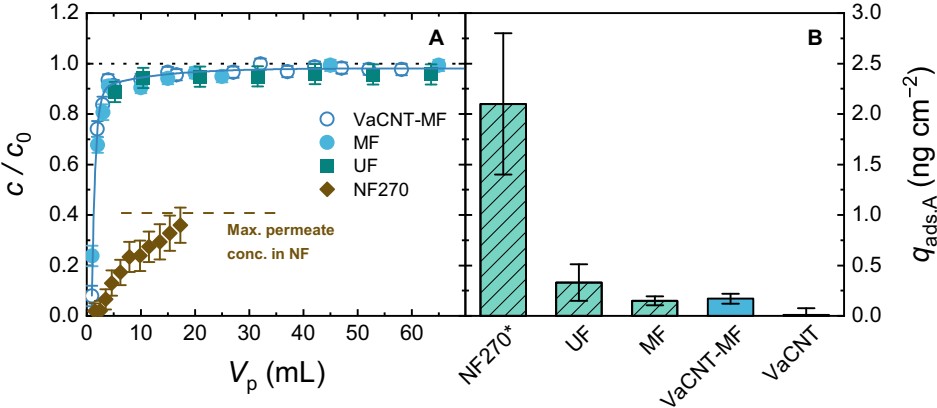

**Fig. 3 | Adsorption property of VaCNT membranes benchmarked against commercial membranes. A** E2 breakthrough (reported as relative permeate concentration $c/c_0$ vs. permeate volume $V_p$) for varied membrane types at the same flux range of 60–100 L m$^{-2}$ h$^{-1}$. **B** Specific E2 adsorbed mass $q_{ads,A}$ at 65 mL except NF270 (permeate volume of 18 mL). The dashed line in **A** indicates the maximum permeate concentration attained due to retention by NF270[25]. The dotted line in **A** indicates where $c/c_0 = 1$. In VaCNT−MF experiment: pore diameter 1.7 nm, 100 ng L$^{-1}$ E2, 1 mM NaHCO$_3$, 10 mM NaCl, pH 8.1 ± 0.2, 23.0 ± 0.2 °C. Error bars represent propagated error from operational parameter variations and analytical error.

diffusion of water would result in additional drag (beyond the reported $F_H$) experienced by the hormone molecules.

## Hormone adsorption in the nanopores
Steroid hormone transport can be detected at high accuracy in small filtration systems by using tritium-labelled molecules, which allows adsorption to be linked with the nanopore properties (hydrophobic, atomically smooth and low-tortuosity pores in VaCNT versus structurally heterogeneous, rough, and tortuous pores in UF/NF). The breakthrough curves in Fig. 3 provide a comparison of hormone adsorption at a relevant flux in UF/NF of 60–100 L m$^{-2}$ h$^{-1}$ between a VaCNT membrane supported by an MF membrane, a UF membrane with a pore diameter of 5.4 nm, and an NF membrane. The results of the UF and NF270 membranes were obtained from Nguyen et al. [76] and Imbrogno and Schäfer[25]. The properties of these membranes are given in Supplementary Table 2.

From Fig. 3A, the relative concentrations of E2 appear the same between VaCNT−MF, MF and UF, which correspond to a similar adsorbed mass of around 0.2–0.3 ng cm$^{-2}$ in Fig. 3B. The E2 adsorbed mass of the VaCNT membrane (determined by subtracting the adsorbed mass of the MF from that of the VaCNT−MF) is insignificant. With the NF270, both the adsorption and retention phenomena occur, and the resulting adsorbed mass was relatively high at 2.3 ± 0.7 ng cm$^{-2}$. This value is similar to those retrieved at higher flux values (up to 300 L m$^{-2}$ h$^{-1}$) and higher permeate volumes, where the adsorption equilibrium at $c/c_0 = 0.4$ was clearly indicated[25]. At a relevant flux in UF/NF membranes, hormone adsorption appears to follow the order NF ≫ UF ≥ MF > VaCNT (negligible). Next, the flow enhancement and adsorption performance will be linked to the VaCNT membrane pore structure.

## Visualisation of the nanopores in VaCNT membranes
Helium ion microscopy was performed to characterise the surfaces and cross-section of the VaCNT membrane (1.7 nm pore diameter). The micrographs are shown in Fig. 4, specifying the top and bottom surfaces that face, respectively, the membrane cell top and the MF support in filtration experiments.

From Fig. 4A, B, both the top and bottom surfaces are porous but these 'pores' (or holes) are large (20–50 nm in diameters in the insets) and do not correspond to the 2 nm diameter carbon nanotube pores. These relatively large holes may provide some extra surface for steroid hormone adsorption, while helium ion microscopy could not resolve the individual nanotube pores.

From a tilt angle (54°) micrograph (Fig. 4C), the top surface of the VaCNT membrane appears rough with many hill and valley areas. Hence, both the hydrophilic functional groups introduced at the membrane surface during membrane fabrication[65], and the spreading of water over the 2–5 μm valley areas contributed to the low contact angle measured with the sessile-drop method (see Supplementary Fig. 3). The good surface wettability may facilitate water and steroid hormone entry in the nanotube pores. It is important to emphasise that the surface characterised using helium ion microscopy and contact angle analysis is different from the pore (carbon nanotube) surface, which is hydrophobic and atomically smooth.

Figure 4D shows the entire cross-section of the VaCNT membrane at a tilt angle, where the vertical alignment of nanotubes can be observed. From the high-resolution micrographs of the cross-section area (Fig. 4E, F), the barrier material (parylene-N) appears to wrap around the carbon nanotubes and form tube-like structures with tube diameters of 5–15 nm. In the same micrographs, the flow paths are highlighted to estimate the pore tortuosity $\alpha$ (which is the ratio between the length of the actual path and that of the hypothetical straight path, see equation (S8) in Supplementary Table 6. The estimated value of $\alpha$ was 1.1, which agrees with the results from the literature of 1.1–1.25[64,65]. The low $\alpha$ implies water flow linearity in the VaCNT membrane and hence few 'collisions' between the wall and adsorbed steroid hormone molecules.

## Breakthrough of hormone (E2) at different fluxes
The dominating flow hydrodynamics may explain the poor adsorption by the VaCNT membrane, as steroid hormone molecules are pushed to exit the nanopores. To examine where adsorption by VaCNT membranes became significant, the flux was then varied between 6 and 60 L m$^{-2}$ h$^{-1}$. The E2 breakthrough curves with VaCNT−MF are shown in Fig. 5, whereas the E2 breakthrough curves with only the MF support is given in Supplementary Fig. 4.

From Fig. 5A, no retention of E2 (with a molecular weight of 272 g mol$^{-1}$ and hydrodynamic diameter of 0.8 nm) was observed because E2 is smaller than the VaCNT membrane pore diameter of 1.7 nm. The removal of smaller species in the feed water matrix, such as electrolytes (Supplementary Fig. 5) and ethanol (Supplementary Fig. 6), was also insignificant. As observed in Fig. 5B, at the high fluxes of 38−60 L m$^{-2}$ h$^{-1}$, the specific E2 adsorbed mass with the VaCNT−MF was similar to that with the MF, which is indicated by the dotted horizontal line (the corresponding breakthrough curves are given in Supplementary Fig. 4) implying negligible E2 adsorption by the VaCNT

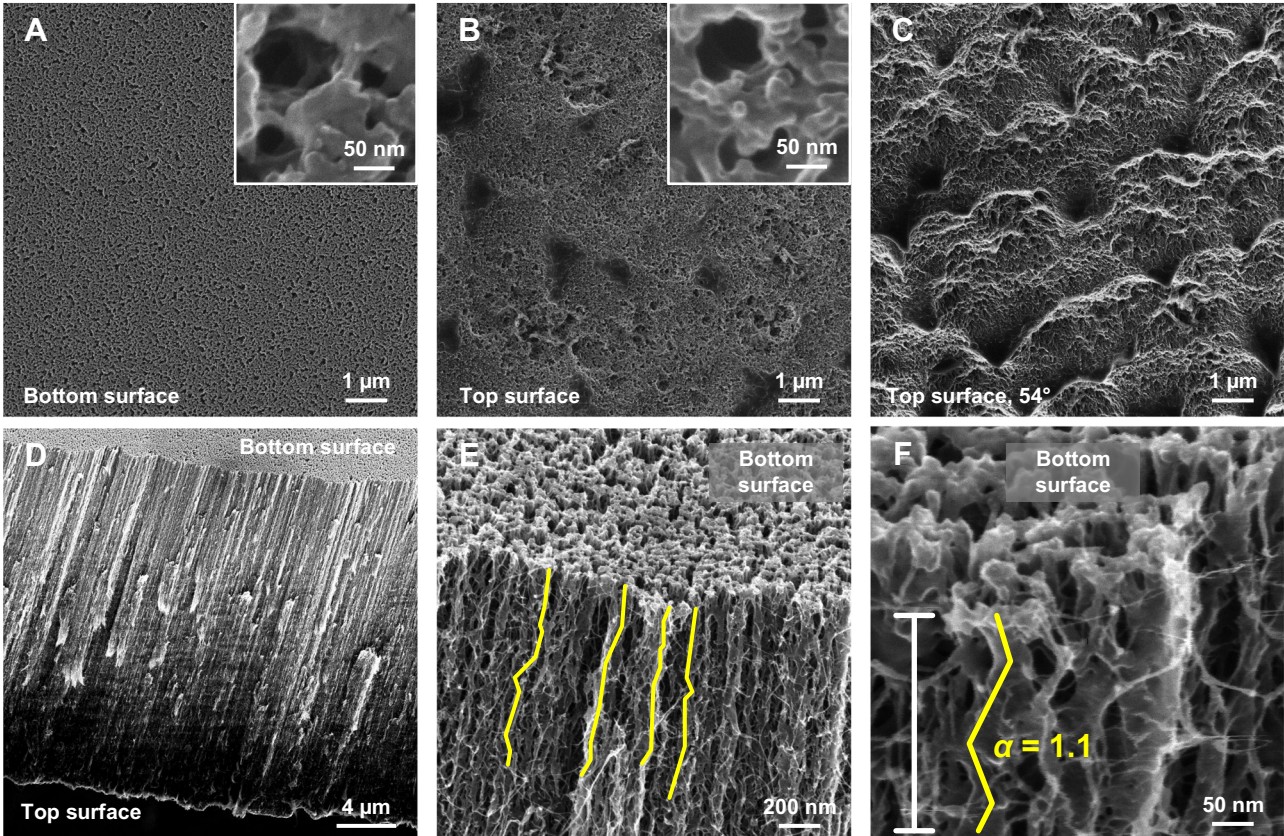

**Fig. 4 | Surface and pore morphology of VaCNT membranes. A, B** Top and bottom surface of the VaCNT membrane (the insets resolve the membrane 'pores' on both surfaces). **C** View of the top surface at a tilt angle of 54° showing the roughness of this surface. **D, E, F** Increasing zooms of the VaCNT membrane cross-section at a tilt angle of 54°. The possible flow paths highlighted in **E** and **F** gives an estimate of the pore tortuosity.

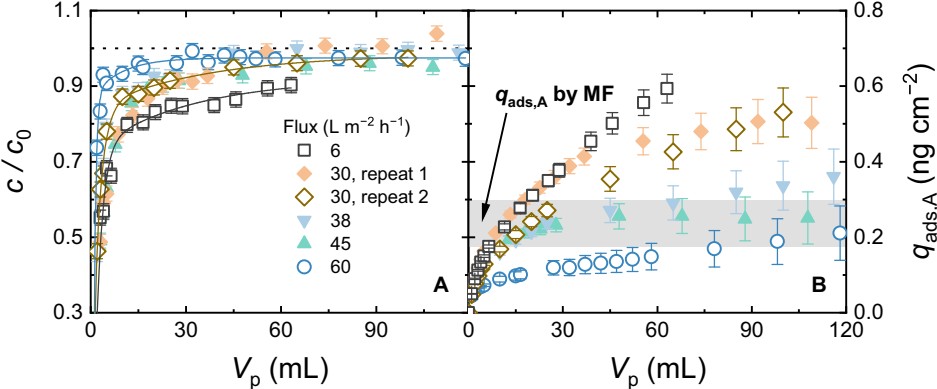

**Fig. 5 | E2 adsorption with VaCNT−MF membranes at different fluxes. A** Breakthrough curves of E2. **B** Specific adsorbed mass $q_{ads,A}$ vs. permeate volume $V_p$. The dotted line in **A** indicates where $c/c_0 = 1$. The dashed line and grey box in **B** indicate the adsorbed mass by MF and error bar at 100 mL. Carbon nanotube pore diameter 1.7 nm, 100 ng $L^{-1}$ E2, 1 mM NaHCO₃, 10 mM NaCl, pH 8.1 ± 0.2, 23.0 ± 0.2 °C. Error bars represent propagated error from operational parameter variations and analytical error.

membrane. E2 adsorption by the VaCNT membrane became significant when the flux was lower at 6–30 L m$^{-2}$ h$^{-1}$.

**Adsorption with varying hydrodynamic drag force**

To quantify adsorption with varying flux, the specific E2 adsorbed mass at 65 mL as a function of flux, flow velocity, and hydrodynamic drag force $F_H$ is reported in Fig. 6.

Figure 6A shows that the specific E2 adsorbed mass of the MF support was 0.22 ± 0.05 ng cm$^{-2}$ and was independent of flux. In contrast, the specific adsorbed mass of VaCNT−MF increased from 0.20 to 0.60 ng cm$^{-2}$ with decreasing flux from 57 to 6 L m$^{-2}$ h$^{-1}$. The relationship between adsorbed mass by the VaCNT membrane and the drag force $F_H$ is illustrated in Fig. 6B. E2 adsorption was significant (above the experimental detection limit) only when the drag force $F_H$ was below 2.2 × 10$^{-3}$ pN corresponding to a flow velocity of 3.0 × 10$^{-4}$ m s$^{-1}$. Decreasing $F_H$ from 2.2 × 10$^{-3}$ to 4.3 × 10$^{-4}$ pN results in an increase in E2 adsorbed mass from zero to 0.40 ng cm$^{-2}$.

To explain the trend in E2 adsorbed mass with the drag force $F_H$, different mass transport processes in the VaCNT membrane pores, namely, diffusion, convection, and adsorption, were evaluated. In

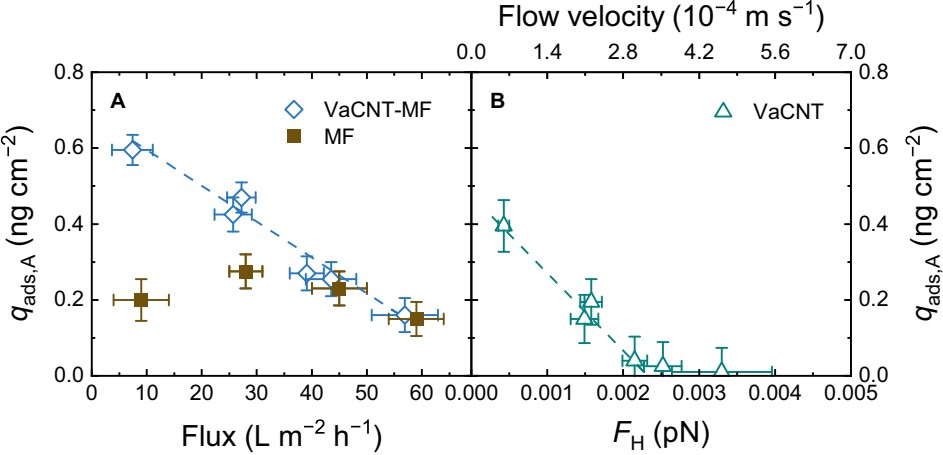

**Fig. 6 | Relationship between hormone adsorbed mass and hydrodynamic drag force $F_H$. A** Specific E2 adsorbed mass $q_{ads,A}$ of VaCNT−MF and MF vs. flux. **B** $q_{ads,A}$ of VaCNT membrane vs. hydrodynamic drag force $F_H$ and flow velocity. Carbon nanotube pore diameter 1.7 nm, permeate volume 65 mL, 100 ng L$^{-1}$ E2, 1 mM NaHCO$_3$, 10 mM NaCl, pH 8.1 ± 0.2, 23.0 ± 0.2 °C. Error bars represent propagated error from operational parameter variations and analytical error.

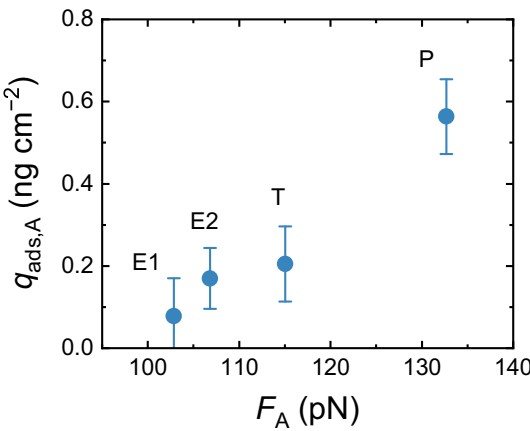

**Fig. 7 | Relationship between hormone adsorbed mass and adhesive force $F_A$.** Specific steroid hormone adsorbed mass of VaCNT membrane $q_{ads,A}$ vs. adhesive van der Waals adhesive force $F_A$. Carbon nanotube pore diameter 1.7 nm, permeate volume 100 mL, flux 27 ± 3 L m$^{-2}$ h$^{-1}$, 100 ng L$^{-1}$ steroid hormone, 1 mM NaHCO$_3$, 10 mM NaCl, pH 8.1 ± 0.2, 23.0 ± 0.2 °C. Error bars represent propagated error from operational parameter variations and analytical error.

nanoconfinement of the VaCNT membrane pores, the E2 is in proximity to the wall, and the diffusion limitation that is relevant in the bulk phase is eliminated[86]. Adsorption (hormone−wall interaction) is fast and occurs within nanoseconds according to molecular dynamics[87]. The adsorption time was much shorter than the hydraulic residence time in VaCNT membranes (between 0.05 and 0.4 s, calculated via equation (S15), Supplementary Table 6). Without the limitations of diffusion and adsorption, the hydrodynamics resulting in the convective flow may determine E2 adsorption. In particular, if $F_H$ overcomes a 'threshold' of 2.2 × 10$^{-3}$ pN, E2 no longer appears to be removed. This threshold may not correspond to the balance between $F_H$ and the hormone−wall friction force $F_F$ (in such a case, the steroid hormone movement is zero). Instead, with $F_H$ below the threshold, the velocity of steroid hormone movement is low enough as such some mass loss is observed, which results in an apparent 'adsorption'. Generally, the magnitude of $F_H$ is related to the type of nanopores. For instance, the no-slip boundary condition inside the pores of commercial UF/NF membranes may result in insignificant or very low drag force $F_H$, whereas in advanced membranes where a partial slip boundary condition exists (such as VaCNT membranes, see Fig. 1), the

magnitude of the drag force $F_H$ will depend on pore chemistry and morphology, as well as on flow conditions.

## Adsorption with varying adhesive force

When the drag force $F_H$ is below the 2.2 × 10$^{-3}$ pN threshold, the strength of the adhesive (van der Waals) force $F_A$ between the hormone and the carbon nanotube wall may determine the amount of adsorbed hormone. To verify whether $F_A$ influences adsorption, the adsorbed masses of four steroid hormone types (E1, E2, T, and P) at the same weak drag force $F_H$ (around 1.6 × 10$^{-3}$ pN) are compared in Fig. 7. The hormone breakthrough curves are given in Supplementary Fig. 7, and the adsorption affinity of the VaCNT membrane is benchmarked against several carbon-based nanoparticles, as shown in Supplementary Fig. 8. The adhesive force is independent of flow velocity, and if a uniform Hamaker constant is applied for the four steroid hormone types, the adhesive force follows the trend in hormone diameter (E1 < E2 < T < P) due to the van der Waals interaction.

The specific adsorbed mass of steroid hormones increases with increasing adhesive force $F_A$ and follows the trend E1 (insignificant) < E2 ≤ T < P. It is implied that a degree of selectivity was achieved. The magnitude of $F_A$ appears to influence the movement of hormone molecules along the nanopores and hence hormone adsorption. The resistance to hormone movement is depends on the strength of the hormone−wall interaction, where the adhesive force $F_A$ impacts the friction force $F_F$ between the hormone and the wall. The hormone molecule can only move when the drag force $F_H$ overcomes this friction force $F_F$. Because the carbon nanotube surface exhibits superlubrication and resists molecule adhesion[88], the friction force $F_F$ is relatively weak and may be comparable in magnitude to the drag force $F_H$. Despite the relatively strong adhesive force $F_A$ (i.e. five orders of magnitude stronger than $F_H$ and $F_F$), adsorption is restricted to a separation distance of -0.3 nm where $F_A$ is countered by the equally strong repulsive force $F_R$ between the electrons of the carbon nanotube wall and those of the steroid hormone. $F_A$ determination is not limited to only the four steroid hormone types but can be applied to other uncharged or charged solutes confined inside various nanopore types. The impact of the geometry and orientation of confined molecules in nanopores on $F_A$ can be inspected computationally, for instance with molecular docking simulations[89].

## Adsorption with varying VaCNT membrane pore diameter

To determine any influence of the pore diameter on steroid hormone adsorption, the E2 adsorbed masses with three different VaCNT membrane pore diameters (1.7, 2.6, and 3.3 nm) are compared in Fig. 8.

The E2 breakthrough curves are given in Supplementary Fig. 9. The flux was controlled at $27 \pm 3$ L m$^{-2}$ h$^{-1}$ during the adsorption experiments.

Under the large slip conditions of this study (see Supplementary Fig. 2) that result in 2.0–3.5 orders of magnitude of enhancement factors (Fig. 8), a plug-like flow with a uniform water velocity profile can be assumed in all cases. With this assumption, the hydrodynamic drag force $F_H$ at the fluid−wall interface scales linearly with the flow velocity according to Eq. (3). The nominal flow velocity (under the assumption that all pores are open to transport) was $(2.2 \pm 0.2) \times 10^{-4}$ m s$^{-1}$ for the VaCNT membranes with pore diameters of 1.7 and 2.6 nm, and $(3.6 \pm 0.1) \times 10^{-4}$ m s$^{-1}$ for the membrane with a pore diameter of 3.3 nm. A modest variation in the drag force $F_H$ was then determined (around 25%). Because the adhesive force $F_A$ at the fluid−wall interface specific for the E2 − carbon nanotube pair is constant (which is around 110 pN), the molecular trajectory did not vary significantly at this pore diameter scale, and it is hypothesised that a small variation in VaCNT pore diameter, between 1.7 and 3.3 nm, did not influence the interplay of forces and hence the adsorbed mass of E2. Consistently with the hypothesis, the specific E2 adsorbed mass

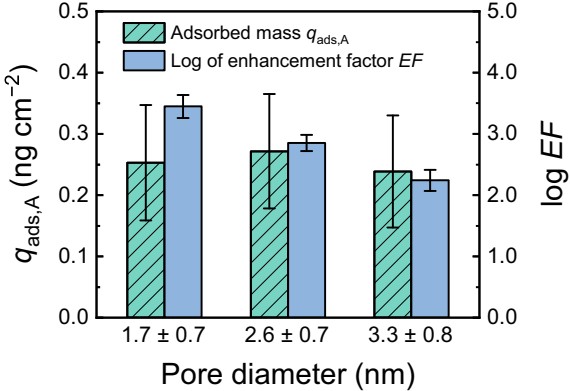

**Fig. 8 | Relationship between hormone adsorbed mass and pore diameter.** Specific E2 adsorbed mass of VaCNT membrane $q_{\mathrm{ads,A}}$ and logarithm of the enhancement factor (EF) with varied membrane pore diameters. Permeate volume 100 mL, flux $27 \pm 3$ L m$^{-2}$ h$^{-1}$, 100 ng L$^{-1}$ E2, 1 mM NaHCO$_3$, 10 mM NaCl, pH $8.1 \pm 0.2$, $23.0 \pm 0.2$ °C. Error bars represent propagated error from operational parameter variations and analytical error.

measured under these conditions was $0.25 \pm 0.10$ ng cm$^{-2}$ and independent of VaCNT membrane pore diameter (Fig. 8).

## Interplay of the forces in nanopores

Based on the above findings for VaCNT membranes with low-tortuosity cylindrical pores, steroid hormone adsorption in various types of nanopores can be explained by the interplay of the hydrodynamic drag ($F_H$), hormone−wall friction ($F_F$), adhesive van der Waals ($F_A$) and repulsive forces ($F_R$), as illustrated in Fig. 9. To examine the trajectory of steroid hormone molecules in the pores, the characteristic diffusion and convection (hydraulic residence) times of NF, UF, MF and VaCNT membranes are given in Supplementary Table 4.

Figure 9A presents an ideal case of a VaCNT membrane with a thickness of 50 μm and pore diameter of 2 nm. With a diffusivity of hormone in the order of $10^{-10}$ m$^2$ s$^{-1}$ (Supplementary Fig. 10), the hormone instantaneously arrives at the pore wall (i.e. within $10^{-11}$ s, see Supplementary Table 4). No adsorption is expected at high fluxes where the plug-like flow condition in the VaCNT membrane applies and results in high $F_H$ and insignificant $F_F$. From experiments with actual VaCNT membranes (Fig. 9B), at lower fluxes that result in weaker $F_H$, an adsorbed steroid hormone amount of up to 0.4 ng cm$^{-2}$ was achieved, suggesting that the transport of hormones in the VaCNT membrane was not entirely frictionless. Some friction may come from the collisions between the steroid hormones and structural defects existing in the carbon nanotube wall (see Fig. 4). Below a flux of 38 L m$^{-2}$ h$^{-1}$, the drag force $F_H$ was not strong enough to overcome the friction force $F_F$ and hence apparent adsorption is observed.

Figure 9C presents the case of an NF270 membrane with an active layer thickness of 0.05 μm and pore diameter of 1 nm. The hormone molecule is always at the pore wall because the pore diameter is similar to the diameter of the hormone. The pore wall roughness, pore hydrophilicity, and pore tortuosity enhance the hormone−wall friction force $F_F$ and decimate the drag force $F_H$ (because the flow velocity is effectively zero with a no-slip boundary condition). Therefore, the hormone is adsorbed at a relatively large amount in these NF membrane pores (e.g. 0.6–1.5 ng cm$^{-2}$ with NF270 membranes[24,25]). The strong friction force $F_F$ along with the weak drag force $F_H$ at the fluid−wall interface can explain the relatively high adsorption of steroid hormone molecules that are confined in NF and UF membranes. The specific affinity between the pore surface and hormone molecules (which is described via the Hamaker constant) affects the magnitudes of the adhesive force $F_A$ and friction force $F_F$, and explains the

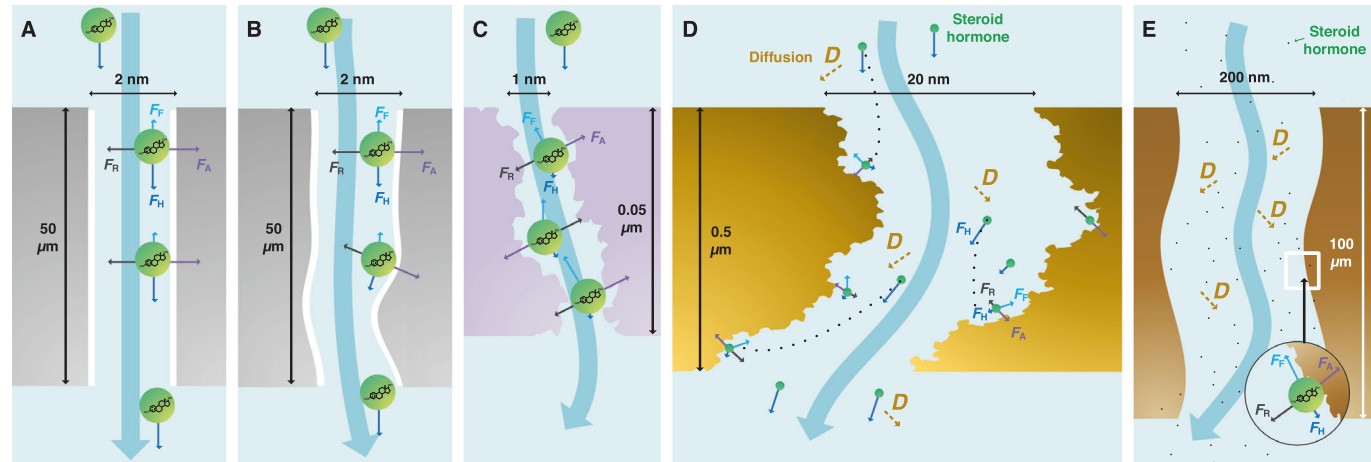

**Fig. 9 | Schematic of the forces (drag force $F_H$, adhesive force $F_A$, friction force $F_F$, and repulsive force $F_R$) acting on the hormone molecule. A** In an ideal case of VaCNT membrane with perfect cylindrical pores (note that the actual membrane is not as perfect), **B** In an actual VaCNT, **C** In an active layer of NF. **D** In a dense layer of

UF. **E** In an MF. The pore and steroid hormone diameters are to scale; the pore lengths and pore diameters are not to scale; and the axial ($F_H$ and $F_F$) and radial ($F_A$ and $F_R$) forces are not to scale with each other.

different adsorbed masses of micropollutants in membranes of the same type. In nanopores with complex morphology (Fig. 9B, C), the force analysis would be complicated by the varying translational and rotational movements of the hormone molecules.

Figure 9D, E present the cases of a loose UF and an MF, where three characteristics of the steroid hormone transport are noted as follows. First, the adhesive force $F_A$ vanishes quickly with increasing hormone–wall distance. From Eq. (4), $F_A$ with a hormone–wall distance $(z - \frac{1}{2} d_{SH})$ of 2 and 5 nm are respectively $10^3$ and $6 \times 10^4$ times weaker than the adhesive force $F_A$ when the hormone is at the fluid–wall interface (i.e. $z - \frac{1}{2} d_{SH} = 0.3$ nm). Therefore, when the hormone molecule is in the pore centre, its radial movement is controlled by diffusion instead of hormone–wall interaction. Second, the diffusion time is sufficiently short (at the magnitudes of $10^{-7}$ to $10^{-5}$ s, see Supplementary Table 4) so that the steroid hormone can reach the pore wall in both membranes within the hydraulic residence times ($10^{-4}$ to $10^{-1}$ s). Finally, in a no-slip boundary condition, the drag force $F_H$ reduces drastically at the wall and enables adsorption. Significant adsorbed masses of steroid hormone by UF are stated in the literature[76,90].

In summary, the VaCNT membranes are ideal to examine hormone adsorption in nanopores through the interplay of the hydrodynamic drag and friction forces acting on the steroid hormone molecules along the flow direction, and the hormone–wall adhesive and repulsive forces perpendicular to the flow direction. Future experimental and modelling efforts will contribute to the understanding of this force interplay by quantifying the precise force magnitudes specific for the solute and nanopore types. VaCNT membranes with sub-nanometre diameter pores (in the NF membrane range) are not yet available but potentially accomplish steroid hormone removal via size exclusion, while the low adsorption capability of these membranes may promise high selectivity.

Another implication of this work is that adsorption in membrane nanopores depends not only on the adsorption surface and mass transfer limitation, which dynamic adsorption studies regard as the only key factors, but also on the interplay of hydrodynamic and adhesion forces. For instance, poor hormone adsorption with VaCNT membranes and relatively good adsorption with UF/NF membranes are explained by this force interplay, but not by surface characteristics or mass transfer limitations. Understanding the underlying mechanisms that dictate adsorption and transport in nanopores can allow the design of better membranes. It is important to mention that the objective of this work involving VaCNT membranes is not attaining effective adsorption performance; this can be accomplished at the external surface of randomly arranged CNT layers. The low adsorption achieved with VaCNT membranes should not be considered as 'negative' result because this is a desirable feature of highly-selective membranes. A prerequisite of such membranes should be minimised interactions and/or collisions between the membrane pore surface and the target micropollutants. On the other hand, an ideal adsorptive membranes should possess abundant accessible surfaces with high affinity for adsorbing micropollutants, and be able to retain these micropollutants at the adsorption sites despite the presence of water flow. These properties can be achieved by the design of the pore geometry and/or inner pore surface structure.

## Methods
### Solution chemistry
Radiolabelled steroid hormones [2,4,6,7-³H] 17β-estradiol (E2, batch 2852571), [1,2,6,7-³H] testosterone (T, batch 2837014), and [1,2,6,7-³H] progesterone (P, batch 2852581) from Perkin Elmer, USA, and [6,7-³H] estrone (E1, batch 210311) from BioTrend, Germany, were supplied as solutions in ethanol. The properties of these hormones are given in Supplementary Table 5. A stock solution ($10\,\mu g\,L^{-1}$) of each SH was prepared by diluting the supplied solution in Milli-Q water (Reference

A + , Merck Millipore, USA). Background electrolyte stock solutions were 5 mM NaHCO₃ (dissolved from analytical-grade 99.7% powder, Bernd Kraft, Germany) and 50 mM NaCl (dissolved from analytical-grade 99.9% (CHROMANORM) powder, VWR Prolabo, Germany). The feed or initial solution was prepared by diluting the stock hormone solution with Milli-Q water and background electrolyte stock solutions, which contained $100\,ng\,L^{-1}$ hormone, 1 mM NaHCO₃, 10 mM NaCl, and $15-30\,mg\,L^{-1}$ ethanol (which is the solvent for the supplied steroid hormones). The molar ratio of hormone: ethanol: water is 1: $10^6$: $10^{11}$. pH was adjusted with 0.1 M NaOH (dissolved from analytical-grade (EMSURE) pellets, Merck Millipore, USA) and 0.1 M HCl (diluted from analytical-grade HCl 37% (ROTIPURAN), Carl Roth, Germany).

### Vertically aligned carbon nanotube (VaCNT) membranes
Three types of VaCNT membranes with average pore diameters of $1.7 \pm 0.7$, $2.6 \pm 0.7$, and $3.3 \pm 0.8$ nm were fabricated via the chemical vapour deposition method[37,91] and schematically described in Supplementary Fig. 11. The carbon nanotubes are >90% single-walled for the forest with the largest average diameter, whereas the percentage of single-walled carbon nanotubes exceed 99% for the other forests. Carbon nanotube diameters (defined as the distance between the nanotube wall centres) were determined from the analysis of an extensive set of transmission electron microscopic (TEM) images (> 200), as reported elsewhere[37]. The VaCNT membrane inner pore diameter was calculated from the measured distances between the wall centres in transmission electron microscopic images as shown in Supplementary Fig. 12. The membrane porosity is calculated with equation (S7), Supplementary Table 6. The theoretical surface area and hormone (E2) adsorption capacity are determined in Supplementary Table 7. The porosity varies between $1.9\% \pm 0.6\%$ (3.3 nm diameter) and $3.4\% \pm 2.0\%$ (1.7 and 2.6 nm pore diameters), and the membrane thickness (or height of VaCNT) varies between $26 \pm 2\,\mu m$ and $69 \pm 1\,\mu m$.

The VaCNT membrane integrity was verified with stringent tests (gas (nitrogen) and liquid (water) permeability, and dye rejection)[37,84]. An MF made from polyvinylidene fluoride (PVDF) with 0.22 μm pores (code GVPP, Millipore, USA) was used as a support for the VaCNT membrane and prevented it from collapsing during the filtration experiment. The MF has very high permeability (> 500 L m⁻² h⁻¹ bar⁻¹) so that the total resistance of VaCNT−MF may not be significantly higher than that of VaCNT membranes. The mass adsorbed with the VaCNT membrane was calculated by subtracting the adsorbed mass with the MF from that with the VACNT−MF at the same flux.

The Drop Shape Analyzer (KRÜSS, Germany) measured the sessile-drop (suitable for characterising hydrophobic surfaces) and captive-bubble contact angles (suitable for hydrophilic surfaces) to confirm the surface properties of the top surface of the VaCNT membrane. Both methods were used because the surface hydrophilicity of the VaCNT membrane was unknown. Applying the sessile-drop method to examine a hydrophilic surface will result in a decreasing contact angle over time because the water drop continuously wets the surface and pores[92]. In the sessile-drop method, a Milli-Q water drop with a volume of 4.5 μL was released from a needle with an inner diameter of 0.51 mm onto the top surface of a dry membrane piece. In the captive-bubble method, the membrane piece was submerged in Milli-Q water for 24 h before characterisation. An air bubble (8 μL) was released from a J-shaped needle with an inner diameter 0.493 mm onto the wetted VaCNT membrane surface. The results are shown in Supplementary Fig. 3.

Helium ion microscopy was performed in a Zeiss Orion NanoFab Helium Ion Microscope (Carl Zeiss Microscopy Deutschland GmbH, Germany) to characterise the morphologies of the VaCNT membrane surface and cross-sections. The cross-sections were prepared by applying in-plane tensile stress to the VaCNT membrane until a rupture occurred. In helium ion microscopy analysis, the helium ion beam

current was set to 0.02–0.3 pA at a beam energy of 25 keV. The samples were mounted in a self-made clamping holder and characterised in the pristine state (without a conductive carbon or noble metal coating). To reduce imaging artefacts when appropriate, charge compensation was performed with an electron flood gun.

## Analytical methods

Steroid hormones were quantified using a Tri-Carb 4910 TR liquid scintillation counter (Packard, USA). The activity was determined in triplicate and correlated to hormone concentration based on the calibration with 0.2, 1, 10, 50 and 100 ng L$^{-1}$ standard solutions. The calibration curves in this work are summarised and the detection limit was verified with previous data for E2 (Supplementary Fig. 13). The detection limit was 0.2 ng L$^{-1}$.

A total organic carbon (TOC) analyser (TOC-L, Shimadzu, Japan) was used in the non-purgeable organic carbon mode to quantify ethanol in feed and permeate. The TOC calibration curves are given in Supplementary Fig. 14. The TOC in feed and permeate samples were diluted 5 times to achieve concentrations below 10 mgC L$^{-1}$ and significantly above the detection limit (0.2 mgC L$^{-1}$) of the instrument.

SenTix 81 and TeraCon 325 electrodes connected with a pH/cond 3320 device (WTW, Germany) were used to respectively measure the pH of the feed, and electrical conductivity of the feed and permeate samples.

## Static adsorption experiments

Static adsorption was performed to determine the adsorbed mass of steroid hormone at equilibrium with the VaCNT membrane flakes. The experimental protocol is described elsewhere[93]. In brief, small pieces of VaCNT membrane flakes (which contains the carbon nanotubes and parylene-N barrier material) with a total mass of 2.5 ± 0.1 mg were mixed with the steroid hormone solution containing 100 ng L$^{-1}$ E2, 1 mM NaHCO$_3$, and 10 mM NaCl, in a 250 mL conical flask. The mixture was shaken at 260 rpm in an incubator shaker (Innova 43 R, New Brunswick Scientific, USA) at a set temperature of 20 °C (thus lower than in filtration experiments of 23 °C). At different time intervals (5, 10, 15, 30, and 45 min; 1, 3, 5, 7, 9, 24, and 26 h), 2.5 mL aliquots of solution samples were taken for analysis. The static adsorption results are shown in Supplementary Fig. 8.

## Diffusion cell experiments

A diffusion experiment was performed to determine the diffusivity of hormone (E2) in the VaCNT membrane. The apparatus used was a Side-Bi-Side (code 5G-00-00-20-50-IO, PermeGear/SES-Analysesysteme, Germany), with a feed/permeate compartment volume of 50 mL and an exposed diffusion membrane area of 3.14 cm$^2$. A water jacket was built in to control the temperature via a water chiller (Minichiller 300 OLÉ, Huber Kältemaschinenbau, Germany) set at 23 °C. Each side of the diffusion cell has two ports. One port on each side was attached to a thermo-coupled conductivity sensor (JUMO BlackLine Lf, JUMO, Germany) to measure the solution temperature and electrical conductivity. These data were acquired every second with two JUMO ecoTRANS Lf 03 modules that were controlled by a LabView 2016 programme version (National Instruments, USA). The VaCNT membrane was mounted between two silicon O-rings (SES-Analysesysteme), with the top surface facing the feed solution. At time zero, the feed and permeate compartments were simultaneously filled with 48 mL of 100 ng L$^{-1}$ E2 solution and 48 mL of Milli-Q water, respectively. An assembly of two magnetic mini stirrers (Rotilabo M3, Carl Roth, Germany) was placed underneath the diffusion cell to stir the feed and permeate solutions at 400 rpm. At different time intervals (0.5, 1, 3, 5, 24, 26, 48, 50, 56, 58 and 72 h), 0.5 mL aliquots of feed and permeate solutions were extracted for steroid hormone analysis. The diffusivity of steroid hormones was estimated via equation (S6), Supplementary Table 6. The membrane boundary layer and the pore

entrance effect cannot be excluded, and hence the pore diffusivity was not accurately determined. A plausible protocol to determine the pore diffusivity in VaCNT membranes is as described elsewhere[84]. Diffusivity results are given in Supplementary Fig. 10.

## Filtration experiments

A schematic of the filtration system with a small filtration area (of only 2 cm$^2$)[79] is given in Fig. 10. Filtration was operated in the dead-end mode with the needle valve (NV in Fig. 10) fully closed. This configuration was decided for UF-type filtration because the VaCNT membrane pore diameter (1.7–3.3 nm) falls into the tight UF range (pore diameters 1.6–3.7 nm)[38]. An HPLC pump (Blue Shadow 80 P, Knauer, Germany) provided a constant feed flow rate. A dampener (code 597-1000-50, Analytical Scientific Instruments, USA) is a part of the system but was disconnected in the study because it potentially causes more error in adsorbed mass. The pressure relief valve (set at 24 bar, SS-4R3A, Swagelok, Germany) relieved the pressure if overpressure occurred, and protected the system. An in-house membrane module with filtration area of 2 cm$^2$ and channel height of 0.7 mm[79] held the VaCNT membrane coupon that was placed on top of a MF coupon. The MF coupon prevented the VaCNT membrane from collapsing during the filtration experiment.

The permeate was automatically separated based on volume with a 16-outlet switching valve (SV, model E1379, Knauer, Germany). The feed bottle and permeate vials were covered with aluminium foil to prevent contamination by dust and reduce evaporation. Permeate mass was measured with a balance (AX822, Ohaus, USA) to calculate flux with equation (S9) in Supplementary Table 6.

Inline feed electrical conductivity C1 and temperature T were measured with a thermocouple electrode (BlackLine Lf, JUMO, Germany). Feed temperature was controlled at 23.0 ± 0.2 °C with a water chiller (Minichiller 300 OLÉ, Huber Kältemaschinenbau, set at 23 °C). Permeate electrical conductivity C2 was acquired with a sensor (ET131 headstage connected to an ER825 detector, eDAQ, USA). Pressure transducers P1 and P2 acquired the feed and permeate pressures (range 0–40 bar, model A-10, WIKA, Germany), respectively. Permeability was calculated from the flux and pressure drop with equation (S10) in Supplementary Table 6. System control and experimental data acquisition were done with LabView 2016 programme (version 16.0.0).

The filtration protocol is summarised in Supplementary Table 8. Steroid hormone adsorption by the filtration system (with no membrane) was quantified as shown in Supplementary Fig. 15. Because no dampener was used, the pressure fluctuated to some extent due to pump pulsation. This pressure fluctuation is characterised by the high resolution (every second) pressure data, as shown in Supplementary Figs. 16 and 17. The pressure fluctuation results in a fluctuation in water flux / flow velocity and hence the magnitude of drag force.

During long filtration experiments of 15–50 h, an increase in pressure was observed with most 1.7–2.6 nm pore diameter VaCNT membranes, which suggests a permeability loss of up to 95%. Several mechanisms may be the cause of this: membrane compression, pore blocking caused by contaminants, adsorbed solutes, hormone–ethanol clustering (because of the strong interaction between steroid hormones and ethanol evidenced by the high solubilities of hormones in ethanol, see Supplementary Table 5), and serious constriction (buckling) of carbon nanotubes (Supplementary Fig. 18). An investigation of this issue shows that membrane compression can occur after the first few hours of filtration, although pore blocking mechanisms cannot be quantified and ruled out (see Supplementary Figs. 19 and 20). The cross-section of a VaCNT membrane piece after a filtration experiment was visualised via helium ion microscopy (Supplementary Fig. 21), verifying that membrane compression (if any) did not significantly alter the membrane structure.

Because VaCNT membranes provide anomalous flow enhancement, the enhancement factor (determined from equation (S13),

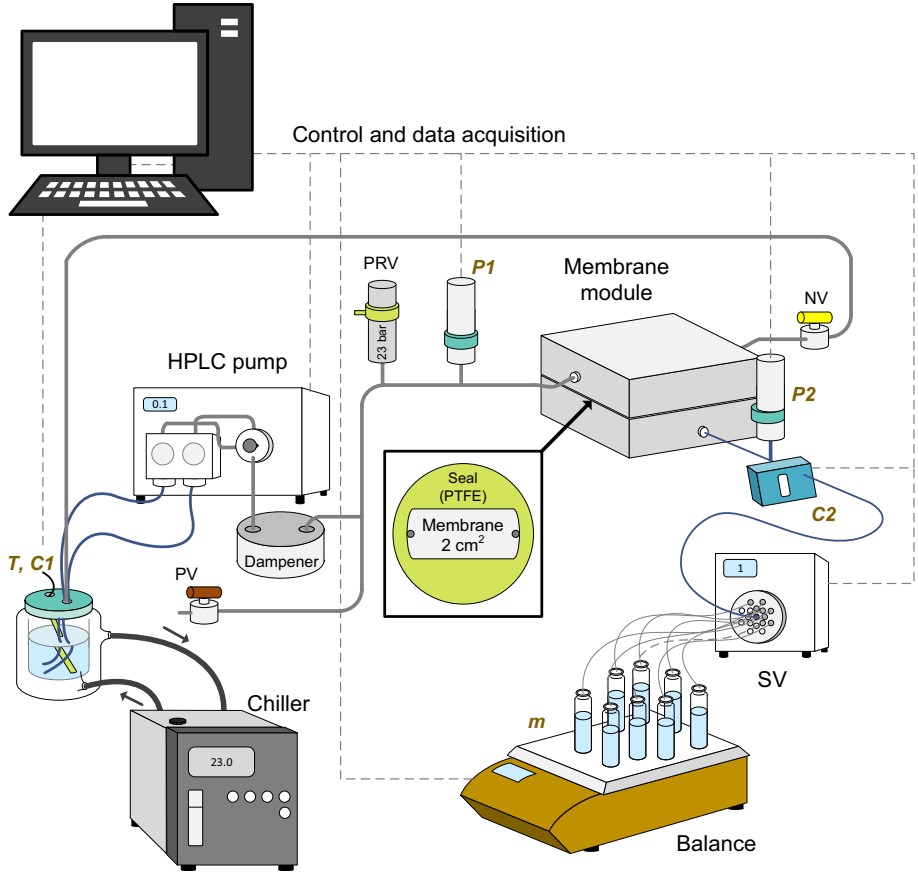

**Fig. 10 | Schematic of the filtration set-up.** PRV pressure relief valve, NV needle valve, SV switching valve, PV purge valve (which, when open, allows system purging with synthetic air). *P1* & *P2*, *C1* & *C2*, and *T* indicate the pressure sensors (feed and permeate), conductivity sensors (feed and permeate), and the temperature sensor (feed), respectively. The PRV relieves the pressure in the case of overpressure (at 24 bar) and protects the system. The dampener is shown in the schematic but disconnected in the filtration experiments.

Supplementary Table 6) is still around 100–1000 despite the substantial decrease in experimental permeability. Hence, the slip condition in VaCNT membranes still appears relevant; the slip velocity and drag force are assumed to depend on the controlled flux and not on the pressure variation.

### Calculations

All equations are given in Supplementary Table 6 and Supplementary Table 7. The assumptions for calculating the hydrodynamic drag, adhesive (van der Waals) and hormone–wall friction forces are shown in Supplementary Discussion 1.

## Data availability

The data that supports the findings of the study are included in the main text and supplementary information files. Raw data can be obtained from the corresponding author upon request. Source data for figures are provided in this paper. Source data are provided with this paper.

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

## Acknowledgements

Helmholtz Recruitment Initiative, Science & Technology of Nanosystems (STN) NanoMembrane, and Helmholtz ERC Recognition Award NAME-PORED are acknowledged for IAMT lab and project funding. BMBF-DAAD provided M.N.N. a PhD research scholarship (NaWaM programme). F.F., M.L.J., and S.F.B. received financial support from the Chemical and Biological Technologies Department of the Defense Threat Reduction Agency (DTRA-CB) *via* grant BA12PHM123 in the "Dynamic Multifunctional Materials for a Second Skin D[MS][2]" programme. Work at LLNL was performed under the auspices of the US Department of Energy under contract DE-AC52-07NA27344. A portion of this work was performed at the Molecular Foundry, which was supported by the Office of Science, Office of Basic Energy Sciences, of the US Department of Energy under contract DE-AC02-05CH11231. F.V. and S.C. (INAM) acknowledge the funding from the Bavarian Ministry of Economic Affairs, Regional Development and Energy (StMWi) under grant 07 02/686 57 and from Oberfrankenstiftung, grant FB00325. The authors thank Peter Weidler (IFG, KIT) for the discussions about the surface area of VaCNTs; Lydéric Bocquet and Marie-Laure Bocquet (ENS/CNRS) for the critical comments on the transport and adsorption in confined spaces and at the graphitic surfaces; and Yoav Green (BGU), Viatcheslav Freger and Uri Sivan (Technion), and Menachem Elimelech (Yale University) for the helpful comments on the forces.

## Author contributions

A.I.S. and F.F. conceived the project and provided expertise in membranes and transport in carbon nanotubes, respectively. M.N.N. and A.I.S. developed the concept of this work. M.N.N. designed and conducted all experiments with the VaCNT membranes. S.J.P synthesised and characterised the CNT forests by transmission electron microscopy. Membrane fabrication and characterisation were performed by M.L.J., S.F.B, and F.F., including measurements of water and gas permeability, integrity tests, and imaging by scanning electron microscopy. F.V. and S.C. conducted helium ion microscopy characterisation on membrane samples. M.N.N. wrote the manuscript. All authors have revised the manuscript.

## Funding

## Competing interests

The authors declare no competing interests.
