## [Peer Review File · Nature Communications]

Interplay of the forces governing steroid hormone
micropollutant adsorption in vertically-aligned carbon
nanotube membrane nanoporesEditorial Note: This manuscript has been previously reviewed at another journal that is not operating a transparent peer review scheme. This document only contains reviewer comments and rebuttal letters for versions considered at *Nature Communications*. Mentions of prior referee reports have been redacted.

REVIEWERS' COMMENTS

Reviewer #1 (Remarks to the Author):

[REDACTED]. I am in favor of publication of the revised manuscript.

Reviewer #3 (Remarks to the Author):

the revision looks fine.

Reviewer #4 (Remarks to the Author):

In order to design better membranes, it is necessary to understand the mechanisms of adsorption and transport in nanopores. This work looks like a mechanistic study. The data obtained were analyzed in a systematic and rational manner. Membranes are mainly used for adsorption or retention of pollutants, but this work does not provide even basic retention or adsorption data. The innovation of the paper is not sufficient, so I cannot recommend its publication in Nature Communications.

REVIEWERS' COMMENTS

Reviewer #1 (Remarks to the Author):

[REDACTED]. I am in favor of publication of the revised manuscript.

Thank you very much for your positive conclusion

Reviewer #3 (Remarks to the Author):

the revision looks fine.

Thank you very much!

Reviewer #4 (Remarks to the Author):

In order to design better membranes, it is necessary to understand the mechanisms of adsorption and transport in nanopores. This work looks like a mechanistic study. The data obtained were analyzed in a systematic and rational manner. Membranes are mainly used for adsorption or retention of pollutants, but this work does not provide even basic retention or adsorption data. The innovation of the paper is not sufficient, so I cannot recommend its publication in Nature Communications.

Thank you very much for your comments that the work is indeed a mechanistic study - and to us and membrane community this work conveys important knowledge. The data were analysed in a systematic and rational manner. Concerning the innovation not being sufficient, there is no other data available that covers this topic. If we are wrong, we would like to see it. If anything, we are probably ahead of current knowledge, based on the challenges we faced when discussing this work with very established colleagues, which is what an excellent journal ought to publish.

As for your concerns about basic retention or adsorption data – and the statement ‘that the work does not even basic retention or adsorption data’ is outright wrong. Below we repeat what is already provided in the manuscript and supporting information in terms of (more than basic) retention and adsorption data. We trust that this will satisfy editor and reviewer.

Figure 5 in the main text shows the breakthrough curves at different fluxes, where apart from at the lowest flux (6 L/m².h), the permeate concentration reached the value of the feed concentration within 65 mL, which indicates that the adsorption saturation is reached and there is no retention (otherwise the permeate concentration would be lower than the feed concentration). Figure 5 also shows the adsorbed masses of E2 as functions of permeate volume, which should qualify as adsorption data. The VaCNT membranes did not retain 17 β -estradiol (E2) in all experiments. This is because the hydrodynamic diameter of the hormone molecule is smaller than the pore diameter (1.7–3.3 nm).

Figure 5 and relevant discussions are shown in lines 299–315, pages 12–13 as follows.

“The E2 breakthrough curves with VaCNT–MF are shown in Figure 5, whereas the E2 breakthrough curves with only the MF support is given in Figure S4.

Figure 5. Relative permeate concentration c / c_0 (A) and specific adsorbed mass $q_{ads,A}$ (B) vs. permeate volume V_p with VaCNT–MF membrane at varied fluxes. The dashed line and grey box in B indicate the adsorbed mass by MF and error bar at 100 mL. Carbon nanotube pore diameter 1.7 nm, 100 ng/L E2, 1 mM NaHCO₃, 10 mM NaCl, pH 8.1 ± 0.2, 23.0 ± 0.2 °C.

From Figure 5 A, no retention of E2 (with a molecular weight of 272 g/mol and hydrodynamic diameter of 0.8 nm) was observed because E2 is smaller than the VaCNT membrane pore diameter of 1.7 nm [...] As observed in Figure 5 B, at the high fluxes of 38–60 L/m².h, the specific E2 adsorbed mass with the VaCNT–MF was similar to that with the MF, which is indicated by the dotted horizontal line (the corresponding breakthrough curves are given in Figure S4) implying negligible E2 adsorption by the VaCNT membrane. E2 adsorption by the VaCNT membrane became significant when the flux was lower at 6–30 L/m².h.”

Similar breakthrough curves and graphs of adsorbed mass vs. permeate volume are also presented in the Supporting Information for the support microfiltration membranes (Figure S4), VaCNT membranes for varied steroid hormone types (Figure S7), and VaCNT membranes with varied pore diameters (Figure S9). The Figures and discussions read as follows:

For Figure S4

“E2 breakthrough and adsorbed mass with the MF support membranes at varying fluxes are given in Figure S4. The adsorbed mass of E2 by the VaCNT membranes can be determined by subtracting the VaCNT–MF membrane data from the MF membrane data. System adsorption (without dampener) is shown with the black diamond symbol.

Figure S4. Permeate E2 concentration c_p (A) and specific adsorbed mass (B) of MF support membranes vs. permeate volume at different fluxes. 100 ng/L E2, 1 mM NaHCO₃, 10 mM NaCl, pH 8.1 ± 0.2, 23.0 ± 0.2 °C. Adsorbed mass by

the filtration system (without dampener) was 0.15 ng/cm^2 (the black arrow in A indicate the feed concentration in this test, of 95 ng/L).

Complete breakthrough was achieved with all fluxes where the MF membrane material is saturated with SH and adsorption is no longer significant ¹ (in this case, where the permeate concentration reaches the feed concentration). The adsorption saturation was approached at around 0.22 ng/cm^2 where the permeate volume reached 45 mL in all experiments.

For Figure S7 (note: for progesterone, the permeate concentration did not reach the level of the feed concentration because adsorption was not saturated, rather than because of retention)

“To verify whether a trend in adsorption can be identified with adhesive force, the breakthrough curves and adsorbed masses of the four SHs were compared in Figure S7.

Figure S7. Permeate concentration c_p (A) and specific adsorbed mass $q_{ads,A}$ (B) vs. permeate volume V_p with different SHs. Flux $27 \pm 3 \text{ L/m}^2 \cdot \text{h}$, 100 ng/L E2, 1 mM NaHCO_3 , 10 mM NaCl , pH 8.1 ± 0.2 , $23.0 \pm 0.2 \text{ }^\circ\text{C}$.

No retention of E1, E2 and T was observed. For P, permeate concentration appeared to be level at 70 ng/L but this may be because of adsorption instead of retention. P (0.86 nm in diameter) is larger than E1, E2, and T ($0.79\text{--}0.82 \text{ nm}$) but smaller than the VaCNT membrane pore diameter ($\sim 1.7 \text{ nm}$).

The adsorption of E1, E2 and T by VaCNT–MF membranes approached saturation after 100 mL . The adsorption of P by both VaCNT–MF and MF membranes did not reach saturation. The adsorbed mass of P by both MF and VACNT–MF membranes was higher than those of E1, E2 and T.

In conclusion, SH adsorption by the VaCNT membranes is specific to the SH type.”

For Figure S9

“To determine whether the VaCNT membrane pore diameter in the nanometre range limits adsorption, the E2 breakthrough curves and adsorbed masses with VaCNT membranes that have three pore diameters ($1.7, 2.6$ and 3.3 nm) were compared in Figure S9.

Figure S1. Permeate E2 concentration c_p (A), adsorbed mass per CNT surface area (B) and specific adsorbed mass (C) of VaCNT–MF membranes with varying VaCNT pore diameter. 100 ng/L E2, 1 mM NaHCO₃, 10 mM NaCl, pH 8.1 ± 0.2, 23.0 ± 0.2 °C.

Complete breakthrough was observed with all pore diameters. Adsorption saturation (where the adsorbed mass no longer increases) was approached at around 100 mL. No trend in the specific adsorbed mass with pore diameter can be determined.

From the trend in adsorbed mass per CNT surface area, it appears that the CNT surface does not limit adsorption. The trend in adsorption kinetics cannot be determined because the data points for different membrane pore diameters overlap.”

The adsorption data have been communicated throughout the manuscript as the specific adsorbed mass of hormones at a particular permeate volume are compared in Figure 6, Figure 7, and Figure 8 of the main text. These Figures read as follows.

Figure 6. Specific E2 adsorbed mass $q_{ads,A}$ of VaCNT–MF and MF vs. flux (A), and $q_{ads,A}$ of VaCNT membrane vs. hydrodynamic drag force F_H and flow velocity (B) at 65 mL permeate volume. The dashed lines are guides for the eye. Carbon nanotube pore diameter 1.7 nm, 100 ng/L E2, 1 mM NaHCO₃, 10 mM NaCl, pH 8.1 ± 0.2, 23.0 ± 0.2 °C.

Figure 7. Specific steroid hormone adsorbed mass of VaCNT membrane $q_{ads,A}$ vs. adhesive van der Waals force F_A . The dashed curve is a guide for the eye. Carbon nanotube pore diameter 1.7 nm , flux $27 \pm 3 \text{ L/m}^2\cdot\text{h}$, 100 ng/L steroid hormone, 1 mM NaHCO_3 , 10 mM NaCl , $\text{pH } 8.1 \pm 0.2$, $23.0 \pm 0.2 \text{ }^\circ\text{C}$.

Figure 8. Specific E2 adsorbed mass of VaCNT membrane $q_{ads,A}$ and logarithm of the enhancement factor (EF). Flux $27 \pm 3 \text{ L/m}^2\cdot\text{h}$, permeate volume 100 mL , 100 ng/L E2, 1 mM NaHCO_3 , 10 mM NaCl , $\text{pH } 8.1 \pm 0.2$, $23.0 \pm 0.2 \text{ }^\circ\text{C}$.

References

- Patel H., Fixed-bed column adsorption study: A comprehensive review, *Appl. Water Sci.*, 9 (2019) 45.